# Influenza Vaccination Coverage and Its Predictors among Self-Reported Diabetic Patients—Findings from the Hungarian Implementation of the European Health Interview Survey

**DOI:** 10.3390/ijerph192316289

**Published:** 2022-12-05

**Authors:** Gergő József Szőllősi, Nguyen Chau Minh, Jenifer Pataki, Cornelia Melinda Santoso, Attila Csaba Nagy, László Kardos

**Affiliations:** Faculty of Health Sciences, University of Debrecen, 4032 Debrecen, Hungary

**Keywords:** influenza vaccination, vaccination coverage, Hungary, diabetes

## Abstract

In high-risk populations, such as the elderly or those with serious medical issues, for instance, people with cardiovascular diseases or diabetes, influenza can have devastating effects because it might contribute to severe complications or even death. This makes vaccination against influenza an essential component of public health. The primary objective of our research was to identify the characteristics that influenced whether an individual chose to become vaccinated against influenza, with an emphasis on whether they reported having diabetes. The data were obtained from the Hungarian implementation of the European Health Interview Surveys, which were conducted in 2009, 2014, and 2019. The total sample size was 15,874 people. To determine the variables that were related to vaccination, a multivariate logistic regression analysis that included interactions was performed. The overall vaccination coverage was 13% in 2009 and 12% in 2014 and 2019 among non-diabetic respondents; the coverage was 26% in 2009, 28% in 2014, and 25% in 2019 among diabetic respondents. Despite vaccination coverage in both groups being below the optimal level of 75%, we were able to identify factors influencing vaccination coverage. Among diabetic respondents, younger age, lower education level, sex, and co-morbidities were factors that influenced vaccination status. It is important for authorities managing healthcare and medical practitioners to be aware of the potential effects that influenza can have on diabetic patients; therefore, more efforts need to be made to increase the number of diabetic people receiving a vaccination against influenza.

## 1. Introduction

Influenza is one of the most important global public health issues. It is considered a contributing factor and one of the main causes of mortality and morbidity, especially among those suffering from high-risk medical conditions due to chronic diseases, such as diabetes [1,2]. This is further exacerbated by the fact that diabetic patients have a higher chance of developing serious flu complications and have at least a twofold increased risk of suffering from cardiovascular diseases [3,4,5]. The public health importance of diabetes is highlighted by the fact that diabetes was ranked ninth on the list of main causes of mortality worldwide in 2019, with an estimated 1.5 million fatalities per year [3,6,7,8]. Therefore, diabetes mellitus represents a growing burden on society, with considerable economical and health-related consequences. People who have diabetes have an increased risk of experiencing more serious consequences after infection with the influenza virus, including a higher likelihood of being hospitalised and admitted to the intensive care unit [8,9,10,11,12,13,14]. The severity of influenza infection is mostly determined by the immune system and the health status of the affected person. Moreover, people with diabetes are more likely to have influenza contribute to their cause of death and, during annual influenza epidemics, the rates of serious complications and death are higher among diabetic patients, which could be prevented by vaccination [13,15,16]. Because of the constant evolution of influenza viruses, annual vaccination against influenza is still the most effective way to prevent the most adverse consequences of infection because vaccination is known to reduce hospitalization among high-risk people [17,18,19]. Therefore, these people should be the main targets of vaccination programs due to their increased risk of influenza complications [19]. The expected price for an out-of-pocket influenza vaccine is between USD 10 and USD 70; however, annual influenza vaccination is clearly recommended and supported in European countries based on the recommendations of the Centers for Disease Control and Prevention for people with diabetes, which are supported by local Hungarian public health authorities because these patients are more susceptible to influenza infection and cardiovascular or respiratory events during flu seasons [13,20,21].

According to the European Centre for Disease Prevention and Control, the vaccination coverage rate is insufficient in the general population and is considered unsatisfactory among high-risk patients in EU/EEA member states. Even though vaccination coverage is low in the general population, it is also considered low in high-risk groups, because the median influenza vaccination coverage among high-risk groups was 44.9% based on seven countries’ data; therefore, an imperative public health task is to increase vaccination coverage rates [22]. To reduce the burden of influenza, it is essential to develop an understanding of the factors that influence vaccine acceptance. Increasing vaccination coverage is an essential effort that requires collaboration across multiple public health disciplines [23,24,25]. In addition, influenza vaccination coverage among patients with chronic conditions, such as diabetes, is considered an effective indicator of a country’s public health preparedness and awareness. A disadvantage of this precept is that it is not mandatory to monitor the administration of vaccines among high-risk groups; therefore, the net health gain due to vaccination cannot be directly monitored.

### Aims

The main objective of this study was to assess and compare vaccination coverage and its influencing factors among self-reported diabetic and non-diabetic respondents based on the Hungarian implementation of the European Health Interview Survey (EHIS) studies of 2009, 2014, and 2019.

## 2. Materials and Methods

### 2.1. Database

The data were obtained from the 2009, 2014, and 2019 Hungarian implementations of the cross-sectional EHIS, which was conducted in order to establish reliable health indicators in the EU Member States. With the survey, it is possible to estimate the population’s health status, lifestyle characteristics, self-care limitations, physical activity, nutrition, health risk behaviors, and healthcare utilization and satisfaction. Data collection was conducted on representative samples using a standardized questionnaire under the supervision of Eurostat. The databases are not available to the public; however, they can be requested from the Hungarian Central Statistical Office, which conducted and supervised the data collection and primary analysis. The EHIS relied on stratified probability samples related to health status indicators of the Hungarian adult population from private households. Because the approach that was utilized during the original data collection remained the same, all three datasets and their respective sets of variables were regarded as being equivalent, and the three datasets were pooled into one. In each of the surveys that were used, the variables of interest were collected using the same method [26]. The databases are considered representative of the Hungarian population [27].

### 2.2. Data

The main dependent variable and the outcome of the logistic regression model, referred to as ‘Influenza vaccine uptake’, was considered positive when subjects had received their last influenza vaccination within one year. The vaccine’s efficacy varies from year to year; therefore, we pooled people together who had never been vaccinated and people who had not been vaccinated against influenza in the given season because, in terms of protection, they were considered unprotected. The independent variables were sex (male/female), age group (18–64 years/65–X), and education level (primary/secondary/tertiary). Hungarian children go to primary school from the age of 6 to 14, usually in grades 1 to 8, which was taken as primary education. After that, they go to secondary school from the age of 14 to 18, where they can also acquire a vocational qualification, which was considered a secondary educational level. The highest level of education referred to as tertiary consisted of a university diploma or equivalent certificate. Answers related to self-reported health status were the following: very bad, bad, and fair, which were pooled as ‘bad’; and good and very good, which were pooled as ‘good’ self-reported health status. Furthermore, we analyzed healthcare-related variables, such as the most recent visit with a doctor (within one year/more than one year ago) and with a medical specialist (within one year/more than one year ago). The presence of co-morbidities, such as diabetes, cardiovascular or cerebrovascular diseases (yes/no; yes implies presence of any one or more of myocardial infarction/coronary heart disease/hypertension/stroke/atrial fibrillation/other heart diseases), musculoskeletal (yes/no; osteoarthritis/rheumatologic/chronic back pain/chronic neck pain/spine problems or deformities), and gastrointestinal diseases (yes/no; stomach ulcer/duodenal ulcer/liver disease/cirrhosis) was also assessed. We used the indicator for the calendar year in which the primary data were gathered (2009/2014/2019). All variables were self-reported.

### 2.3. Statistical Methods

Univariate and multivariate logistic regression analysis was performed to determine the characteristics that influenced the influenza vaccination status, with a special focus on diabetes. We investigated the collinearity and interactions between the explanatory variables and diabetes with vaccination status as the outcome, and eight potential pairs were identified (Table 1; last column). Then, all the variables with their interactions were assessed in combination, and three pairs (age group and education level, year and education level, sex, and last visit with a doctor) were taken into consideration when the multivariate model was established. The final model contained these three interaction terms. The results are presented in the form of adjusted odds ratios (OR) and *p*-values. Statistical analysis was performed using Stata statistical software (version 13.0, Stata Corp., College Station, TX, USA), and a *p*-value less than 0.05 was taken to indicate statistical significance.

## 3. Results

The initial sample size was 16,480. After data cleaning, which involved the exclusion of those respondents who did not have data regarding diabetes status and were below 18 years of age, the total sample size was reduced to 15,874, with 4899 respondents from the 2009 dataset, 5620 respondents from the 2014 dataset, and 5355 participants from the 2019 dataset. Diabetes was self-reported by 426 participants in 2009, 474 in 2014, and 547 in 2019; therefore, the prevalence of the disease was 9% in 2009 and 8% in 2014, and it increased to 10% in 2019. The overall vaccination coverage in the dataset was 13% (*n* = 2132). The vaccination coverage was 15% (*n* = 709) in 2009 and 13% (*n* = 726) in 2014, and it remained at 13% (*n* = 697) in 2019.

### 3.1. Vaccination Coverage among People with Self-Reported Diabetes

No significant association was observed during the years in terms of vaccination coverage. In 2009, of the 426 self-reported diabetic people, 111 (26%) had received the influenza vaccination; in 2014, only 132 of the 474 (28%) self-reported diabetic people received a vaccination, and in 2019, of the 547 self-reported diabetic people, 139 (*n* = 25%) had received a flu shot within one year. Diabetic people aged 65 years or older (*n* = 745) had significantly higher vaccination coverage, with 35% compared to 17% in people aged 18 to 64 years (*n* = 702) (*p* < 0.001). In the unadjusted assessment, sex had no significant association with vaccination in people with diabetes (*p* = 0.186), even though vaccination coverage was higher among males (28%) than females (25%). No significant difference was observed regarding vaccination between respondents with primary education (*n* = 343, vaccination coverage = 25%) and respondents with secondary education (*n* = 892, vaccination coverage = 24%) (*p* = 0.691); however, a significant difference in vaccine coverage was observed between people with tertiary education (*n* = 211, vaccination coverage = 39%) and primary education (*p* = 0.001). The group with ‘good’ self-reported health status (*n* = 949) had a significantly lower proportion of vaccinated members (25%) compared with the group who considered their health status ‘bad’ (*n* = 496; vaccination coverage = 30%) (*p* = 0.034). Diabetics with frequent (last visit: <12 months before survey time) doctor visits (*n* = 1380, vaccination coverage = 27%) did not have significantly higher vaccination coverage compared to people with infrequent (≥12 months) visits (24%; *p* = 0.632). A similar non-significant contrast was observed regarding the last meeting with a specialist. The group of diabetics who frequently visited a doctor (*n* = 1202) had a higher proportion (27%) of vaccinated members compared with the group with infrequent visits (*n* = 244; vaccination coverage = 22%) (*p* = 0.073). In addition, having a co-morbidity, such as cardiovascular or cerebrovascular, musculoskeletal, or gastrointestinal diseases, was significantly associated with vaccination status. Diabetic people suffering from cardiovascular or cerebrovascular diseases had significantly higher vaccination coverage (28%; *n* = 1184) compared with people who did not suffer from these diseases (21%; *n* = 263) (*p* = 0.026). The same associations were observed for musculoskeletal disorders (*n* = 955, 29% vs. *n* = 492, 21%; *p* = 0.001) and gastrointestinal diseases (*n* = 112, 36% vs. *n* = 1332, 26%; *p* = 0.020). All in all, diabetics with co-morbidities had higher vaccination coverage than those without co-morbidities (Table 1).

### 3.2. Interaction between Diabetes and Explanatory Variables

The association between vaccination status and age group (*p* < 0.001), sex (*p* < 0.025), education level (*p* < 0.001), self-perceived health status (*p* < 0.001), last meeting with a doctor (*p* < 0.001), last meeting with a specialist (*p* = 0.005), cardiovascular or cerebrovascular co-morbidities (*p* < 0.001), and musculoskeletal co-morbidities (*p* < 0.001) was significantly heterogeneous across levels of diabetes, based on logistic regression models unadjusted for any other variables (Table 1).

### 3.3. Multivariate Model with Interactions among People with Diabetes

Diabetic people with a secondary education level who belonged to the age group of 65 years or older had significantly higher odds for the outcome of adequate influenza immunization status compared to people aged 18–64 years with the same education level (OR = 3.67; *p* < 0.001). Furthermore, respondents from the 18–64-year age group with secondary education had significantly lower odds of having an influenza vaccination within one year compared with participants in the same age group with primary education (OR = 0.53; *p* = 0.038). Respondents with tertiary education aged 65 years or older had significantly higher odds of having a flu shot within one year compared with respondents aged 18–64 years with the same level of education (OR = 2.41; *p* = 0.005). Participants aged 65 years or older with secondary education had significantly higher odds of having adequate immunization status compared with members of the same age group with primary education (OR = 1.58; *p* = 0.035). The same phenomenon was observed regarding respondents aged 65 years or older who had a tertiary education compared with people with primary education, indicating that elderly diabetics with tertiary education had significantly higher odds of having received a flu shot compared with people with primary education (OR = 2.49; *p* = 0.001). In 2019, diabetic people with tertiary education had 2.65 times higher odds of having received a flu shot compared with diabetics with primary education in the same year, and this difference was significant (OR = 2.65; *p* = 0.002). Male participants who visited their doctor within one year had 39% higher odds of having an influenza vaccination compared with females who visited their doctor within one year (OR = 1.39; *p* = 0.014). However, the male respondents who did not visit their doctor within one year had 74% lower odds of having an adequate vaccination status compared with females (OR = 0.26; *p* = 0.033). In terms of co-morbidities, diabetic people with accompanying musculoskeletal disorders had significantly higher odds of having an influenza vaccination compared with people who did not suffer from these conditions (OR = 1.43; *p* = 0.015) (Table 2).

### 3.4. Multivariate Model with Interactions among People without Diabetes

Non-diabetic respondents aged 65 years or older had significantly higher odds of having adequate immunization status against influenza in different strata regarding education level. Therefore, respondents with tertiary education had 3.31 times higher odds, respondents with secondary education had 3.37 times higher odds, and respondents with primary education had 4.72 times higher odds of having received a flu shot within one year, which was significant (*p* < 0.001). Regarding the interaction of level of education with age group, non-diabetics aged 18–64 years with a higher education level (secondary OR = 1.52; tertiary OR = 2.45) had significantly (secondary *p* = 0.007; tertiary *p* < 0.001) higher odds of being vaccinated against influenza. For the age group of 65 years or older, the aforementioned association was observed in people with tertiary education (OR = 1.72, *p* < 0.001). People with primary education had lower odds of being vaccinated against influenza in the past year; therefore, people who participated in EHIS 2014 had 41% lower odds (OR = 0.59; *p* = 0.022) and people who participated in EHIS 2019 had 38% (OR = 0.62; *p* = 0.001) lower odds of having a flu shot compared with people participating in 2009. This negative trend was observed with secondary education level, where the odds ratio was 0.72 (*p* < 0.001) in 2014 and 0.59 (*p* < 0.001) in 2019 compared with the reference year (2009). Higher education level proved to be a significant factor related to the interaction of the calendar year. In 2009 (secondary OR = 1.35, *p* = 0.019; tertiary OR = 1.43, *p* = 0.032) and 2014 (secondary OR = 1.67, *p* = 0.036; tertiary OR = 3.06, *p* < 0.001), higher education levels increased the odds of being vaccinated compared with primary level. This trend was observed for non-diabetic people in 2019, where participants with tertiary education had significantly higher odds of having received a flu shot within one year compared to people with primary education (*p* < 0.001). Male respondents who visited their doctor at least more than a year ago had significantly higher odds of having received an influenza vaccination by 42% (OR = 1.42; *p* = 0.037). Females with frequent (<12 months) doctor visits had more than two times higher odds of having received a flu shot compared with men who did not visit their doctor regularly (OR = 2.27, *p* < 0.001). However, men who visited a doctor frequently had higher odds of being vaccinated compared with men who did not visit a doctor regularly (OR = 1.76, *p* < 0.001). In addition, meeting with a specialist in the last year seemed to be a protective factor, significantly increasing the odds of having received a flu shot (OR = 1.53; *p* < 0.001). Non-diabetics with good self-reported health status had lower odds of being vaccinated (OR = 0.81; *p* = 0.005). Having cardiovascular or cerebrovascular co-morbidities (OR = 1.68; *p* < 0.001) and musculoskeletal (OR = 1.29; *p* < 0.001) or gastrointestinal diseases (OR = 1.29; p = 0.012) increased the odds of having received a flu shot (Table 2).

### 3.5. Interaction between Diabetes and Other Explanatory Variables

Based on the logistic regression model adjusted for other explanatory variables, significant interaction regarding diabetes and age group with education level (*p* < 0.001) was found. The interaction between diabetes and education level with the calendar year of the study was not significant (*p* = 0.053). However, diabetes was significantly associated with cardiovascular diseases (*p* = 0.019), but musculoskeletal diseases showed no significant interaction related to diabetes (*p* = 0.517). No significant interactions could be seen for gastrointestinal diseases (*p* = 0.478) and self-reported health status (*p* = 0.708) with diabetes. Diabetes regarding sex and the last visit with a doctor was a significant interaction (*p* = 0.001) in terms of vaccination, but diabetes showed no association with the last visit with a specialist (*p* = 0.370). Sex differences and the last visit with a doctor were found to be significantly (*p* = 0.010) associated with diabetes, but diabetes was not related to educational level during the calendar year (*p* = 0.634). 

## 4. Discussion

In the current study, the EHIS database was used to conduct an analysis of influenza vaccine uptake and its determining factors, focusing on the Hungarian diabetic population. Furthermore, we analyzed the temporal trend from 2009 to 2019 in Hungary, and our findings suggested that influenza vaccination uptake among persons with diabetes did not improve significantly; it was 26% in 2009 and, aside from a non-significant increase in 2014, it decreased to 25% in 2019. This observation is aggravated by the fact that influenza vaccination is provided at no cost to diabetic patients in Hungary [28]. In addition, influenza vaccination is endorsed by public health authorities. The vaccination coverage in Hungary among patients with diabetes was very low, and it was substantially lower than the rate of 75% recommended by the European Union [23]. Although the vaccination coverage was higher among people with diabetes than people without diabetes, it remained below the optimal level. These findings were in line with those found in the international literature [29,30]. In addition, we were able to identify the most important factors that influenced vaccination among Hungarian diabetic people. One of the most influential variables that were associated with vaccination was the education level in both the diabetic and non-diabetic groups. Higher education level contributing to vaccination was in line with the literature [31,32]. Male vs. female differences were observed in the terms of doctor visits and vaccine uptake, indicating that diabetic men with more frequent doctor visits were more willing to be vaccinated, and diabetic men who did not visit their doctor regularly had lower odds of being vaccinated. In contrast, non-diabetic women who did not visit their doctor within one year had increased odds of being vaccinated. This could indicate that differences between males and females may contribute to vaccine acceptance [33]. Furthermore, frequent doctor visits might be associated with greater exposure to vaccination advice; higher education might be associated with higher levels of health literacy which, in turn, might be associated with better vaccine acceptance. Another important factor that we identified was older age, which had a significant effect on vaccination. The elderly population had higher a proportion of vaccinated members, and were thus more willing to be vaccinated; moreover, these individuals are in the care of healthcare providers [34,35,36]. This does not mean that older age increases vaccination coverage, but it does highlight the fact that it would be useful to identify the factors that contribute significantly to vaccination uptake among elderly people. Therefore, it would be important to examine whether public health interventions could be taken to further increase vaccination coverage among the elderly and to raise vaccination coverage in younger age groups to a level similar to that of the elderly. However, this is not within the scope of the present study, but a potential future research pathway that can build substantially on our current findings. In terms of co-morbidities, more serious conditions and their consequences influenced influenza vaccination; people with a more severe co-morbid status needed and had higher vaccine uptake [13,37,38,39].

There is an ongoing debate among the public over the efficacy, safety, and unavoidable nature of influenza vaccinations. This is one theory to explain the lack of progress seen in recent years regarding vaccination rates [18,32,36]. This issue is worsened by the absence of specialized vaccination centers in Hungary; even though such facilities could contribute to increasing vaccine coverage, they do not exist. Instead, most vaccinations are administered at the level of primary care.

Vaccination could be evaluated by vaccination coverage indicators, and the establishment of these indicators would be feasible because the data are available in many cases from the healthcare sector. Therefore, it is important for authorities in charge of healthcare and medical practitioners in nations with a high diabetes burden to be aware of the potential effects that influenza could have on diabetic patients. 

### Strengths and Limitations

Even though the primary objective of EHIS was not to determine the reasons for low vaccine coverage among different strata, our analysis was able to identify the most important determinants of influenza vaccination status. However, using self-report-based questionnaires (e.g., the registered co-morbidities, such as diabetes), may have resulted in under-representation in our results, so it is important to consider these factors. Because of the methodological nature of the data collection, the database contained only information on those who responded. Since there was no information on those who did not respond, it was not possible to evaluate any potential systematic differences between those who responded and those who did not respond.

## 5. Conclusions

One of the most important discoveries we made in our research was that the percentage of diabetic adults in Hungary who receive vaccination against influenza remains below the recommended level, which should have been considered optimal, and this trend did not improve between 2009 and 2019. Therefore, more efforts need to be made to increase the number of people in this high-risk category who receive the flu vaccine. This is especially important for people with a lower education level, those aged 18–64 years, and those who do not have any accompanying co-morbidities that put them at an even greater risk of complications.

## Figures and Tables

**Table 1 ijerph-19-16289-t001:** Descriptive statistics by vaccination coverage of diabetic survey respondents.

		Diabetes: No *n* = 14,427			Diabetes: Yes *n* = 1447			*p*-Value for Heterogeneity across Diabetes Strata
Factor	Level	Stratum Sample Size	Vaccination Coverage (%)	Stratum-Specific *p*-Value	Stratum Sample Size	Vaccination Coverage (%)	Stratum-Specific *p*-Value	
Year	2009	4473	13%		426	26%		0.282
	2014	5146	12%	0.007	474	28%	0.546	
	2019	4808	12%	0.010	547	25%	0.819	
Age group	18–64	11,295	7%		702	17%		<0.001
	65–X	3132	29%	<0.001	745	35%	<0.001	
Sex	Male	6579	11%		674	28%		0.025
	Female	7848	13%	0.009	773	25%	0.186	
Education level	Primary	2173	15%		343	25%		0.001
	Secondary	9226	11%	<0.001	892	24%	0.691	
	Tertiary	3023	14%	0.241	211	39%	0.001	
Self-perceived health status	Good	12,854	11%		949	25%		<0.001
	Bad	1555	24%	<0.001	496	30%	0.034	
Last meeting with a doctor	≥12 months	3522	5%		67	24%		<0.001
	<12 months	10,860	15%	<0.001	1 380	27%	0.632	
Last meeting with a specialist	≥12 months	5745	8%		244	22%		0.005
	<12 months	8588	15%	<0.001	1202	27%	0.073	
Co-morbidities: Cardiovascular or cerebrovascular disease (s)	No	9227	7%		263	21%		<0.001
	Yes	5199	21%	<0.001	1184	28%	0.026	
Co-morbidities: Musculoskeletal disorder (s)	No	8263	8%		492	21%		<0.001
	Yes	6164	18%	<0.001	955	29%	0.001	
Co-morbidities: Gastrointestinal disease (s)	No	13,649	12%		1332	26%		0.244
	Yes	776	22%	<0.001	112	36%	0.020	

**Table 2 ijerph-19-16289-t002:** Factors that influenced influenza vaccination among diabetic and non-diabetic respondents in Hungary based on a multivariate logistic regression model.

Factor (Stratum, If Any)		Diabetes: No		Diabetes: Yes	
	Level	OR	*p*-Value	OR	*p*-Value
Age group (Education level: Primary)	18–64				
	65–X	4.72	<0.001	1.23	0.470
Age group (Education level: Secondary)	18–64				
	65–X	3.37	<0.001	3.67	<0.001
Age group (Education level: Tertiary)	18–64				
	65–X	3.31	<0.001	2.41	0.005
Education level (Age group: 18–64 years)	Primary				
	Secondary	1.52	0.007	0.53	0.038
	Tertiary	2.45	<0.001	1.27	0.521
Education level (Age group: 65–X years)	Primary				
	Secondary	1.88	0.125	1.58	0.035
	Tertiary	1.72	<0.001	2.49	0.001
Year (Education level: Primary)	2009				
	2014	0.59	0.022	1.17	0.743
	2019	0.62	0.001	0.73	0.264
Year (Education level: Secondary)	2009				
	2014	0.72	<0.001	1.00	0.997
	2019	0.59	<0.001	0.73	0.184
Year (Education level: Tertiary)	2009				
	2014	1.25	0.134	1.44	0.402
	2019	1.09	0.571	1.53	0.296
Education level (Year: 2009)	Primary				
	Secondary	1.35	0.019	0.98	0.930
	Tertiary	1.43	0.032	1.26	0.557
Education level (Year: 2014)	Primary				
	Secondary	1.67	0.036	0.84	0.701
	Tertiary	3.06	<0.001	1.57	0.387
Education level (Year: 2019)	Primary				
	Secondary	1.30	0.087	0.98	0.939
	Tertiary	2.52	<0.001	2.65	0.002
Sex (Last meeting with a doctor: <12 months)	female				
	male	1.10	0.105	1.39	0.014
Sex (Last meeting with a doctor: ≥12 months)	female				
	male	1.42	0.037	0.26	0.033
Last meeting with a doctor (Sex: Female)	<12 months	2.27	<0.001	0.46	0.067
	≥12 months				
Last meeting with a doctor (Sex: Male)	<12 months	1.76	<0.001	2.51	0.068
	≥12 months				
Last meeting with a specialist	<12 months	1.53	<0.001	1.29	0.157
	≥12 months				
Self-perceived health status	Good	0.81	0.005	0.85	0.265
	Bad				
Co-morbidities: Cardiovascular or cerebrovascular disease (s)	Yes	1.68	<0.001	1.07	0.699
	No				
Co-morbidities: Musculoskeletal disorder (s)	Yes	1.29	<0.001	1.43	0.015
	No				
Co-morbidities: Gastrointestinal disease (s)	Yes	1.29	0.012	1.54	0.057
	No				

## Data Availability

The data presented in this study are not available to the public but can be requested from the institution that performed and supervised the data collection and primary analysis: Hungary’s Central Statistical Office.

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
