# Peer review of "Influenza Vaccination Coverage and Its Predictors among Self-Reported Diabetic Patients—Findings from the Hungarian Implementation of the European Health Interview Survey"

_ijerph, 2022, doi:10.3390/ijerph192316289_

Round 1

Reviewer 1 Report (New Reviewer)

This research gives a good overview of an important public health issue.
However certain aspects need attention which are as follows:

Why this research excluded less than 18 years aged diabetic patients?

Why were 18-64 years and more than 65 years age groups formulated for this research?

This research needs to define primary, secondary and tertiary education as these terms are not uniform universally.

There is a perception in this manuscript (in lines 11-12 and 37) that Influenza is a cardiovascular problem which of course is not a case. However, these mentions should be preceded by the fact that there has been found associations between influenza and CVS diseases backed by appropriate citation/s. Otherwise, these sentences (in lines 11-12 and 37) are misleading.

Lines 30-2; "It is considered....., especially diabetes" should be appropriately cited.
Lines 33-4; time and place is missing in this sentence, should be mentioned. Also in this sentence the citations given do not conform to the statement, so should be revised.

Lines 45-9; citation must be done here

In Methodology, Lines 87-9 and 89-90 should be cited as the verification of this fact will help clear doubts in the minds of readers.

Author Response

Dear Reviewer, we would like to thank you for giving us the opportunity to submit a revised draft of our manuscript.

Why this research excluded less than 18 years aged diabetic patients?

The 2009 and 2014 EHIS databases contained data regarding the health status of the adult Hungarian population (18+). However, the 2019 database contained data regarding the population under 18 years of age, but the main profile of these data was not relevant to the subject of the manuscript.

Why were 18-64 years and more than 65 years age groups formulated for this research?

The institution that performed and supervised the data collection and primary analysis: Central Statistical Office (Hungary).

During the data collection the Hungarian Central Statistical Office collected data related to the year of birth and age of the respondents, but due to the current GDP regulations, only very broad age group data were available for analysis purpose.

This research needs to define primary, secondary and tertiary education as these terms are not uniform universally.

Correction and extension have been made. Now thy can be read:

The independent variables were sex (male/female), age group (18–64 years/65–X) and education level (primary/secondary/tertiary). Hungarian children go to primary school from the age of 6 to 14, usually in grades 1 to 8, which was taken as primary education. After that, they go to secondary school from the age of 14 to 18, where they can also acquire a vocational qualification, which was considered as secondary educational level. The highest level of education referred to as tertiary consisted of a university diploma or equivalent certificate.

There is a perception in this manuscript (in lines 11-12 and 37) that Influenza is a cardiovascular problem which of course is not a case. However, these mentions should be preceded by the fact that there has been found associations between influenza and CVS diseases backed by appropriate citation/s. Otherwise, these sentences (in lines 11-12 and 37) are misleading.

Corrections have been made in the Abstract and in the Introduction to increase the understanding of the association between diabetes, cardiovascular diseases and vaccination.

Lines 30-2; "It is considered....., especially diabetes" should be appropriately cited.

The document has been enriched with more references.

Lines 33-4; time and place is missing in this sentence, should be mentioned. Also in this sentence the citations given do not conform to the statement, so should be revised.

Corrections have been made. We rephrased the affected sentences for better consistency.

Lines 45-9; citation must be done here

The document has been enriched with new references and the citations have been upgraded.

In Methodology, Lines 87-9 and 89-90 should be cited as the verification of this fact will help clear doubts in the minds of readers.

The same data collection and methodology was used in the EHIS 2009, EHIS 2014 and EHIS 2019. The Central Statistical Office was responsible for the task being carried out, under the supervision of EUROSTAT. All three databases are considered to be representative of the Hungarian population and therefore the same questions of the representative databases were pooled together. Literature references have been added to the Materials and methods to make the chapter more clear and understandable.

Reviewer 2 Report (Previous Reviewer 1)

Authors have provided the justifications and improved the manuscript according to previous comments.

Author Response

Dear Reviewer, we would like to thank you for giving us the opportunity to submit a revised draft of our manuscript.

Reviewer 3 Report (New Reviewer)

This is a manuscript aimed at investigating the influenza vaccination coverage, its predictors among self- reported diabetic patients.

The paper applied the same methodology and data source used in your manuscript “An Exploratory Assessment of Factors with Which Influenza Vaccine Uptake Is Associated in Hungarian Adults 65 Years Old and Older” where the rationale of the study is more solid respect to this new paper.

Indeed, the study presents several criticisms linked to the choice to compare two populations: the self reported individuals with diabetes and that who respond to be non-diabetic and the risk is that the results could be biased by confounding factors that the authors do not take into account.

They authors should consider to adjust for age, sex and comorbities.

The comorbidities were present in both investigated groups? 

Minor comments

Lines 38-39 is to replace after line 41.

Line 47 explain why the effectiveness can be lower than the expected.

A reference for the sentence “The expected price is for an out-of-pocket influenza vaccine is 51 between $20 and $70” is to added.

After reference 10, line 56, you have to begin a new line.

The sentence 59-61 is to rewrite as it is not very clear.

From line 63 the authors should add proper references.

The sentence at line 701 could be deleted.

The study aim is not clear, explain the rationale to compare influenza vaccination coverage between self reported diabetic and non diabetic when non diabetic could suffer other co-morbidities.

The authors can expand the study limits:

-          Use of questionnaire

-          Undiagnosed diabetes among individuals who respond not to be diabetics. It is well known that a considerable percentage of population does not know to be diabetic.

Author Response

They authors should consider to adjust for age, sex and comorbities.

It is extremely important for epidemiological analyses to adjust for several explanatory factors. That is why we adjusted our analysis for confounding factors in the multiple analysis.

The manuscript contains two tables. The first table describes the characteristics of diabetic and non-diabetic respondents, with details on age group (18-64, 65-X), gender (Male, Female) and co-morbidities (Cardiovascular or cerebrovascular disease(s), Musculoskeletal disorder(s), Gastrointestinal disease(s)). The distribution of respondents expressed as row percentages is shown based on the self-reported disease categories.

The second table describes the results of the multiple logistic regression model, which was adjusted for several explanatory variables, whereas possible interactions have been taken into account to obtain more precise relationships, as described in the 2.3 Statistical methods section. This means that the analysis was adjusted for age, sex and comorbidities and other factors.

The comorbidities were present in both investigated groups? 

The descriptive analyses in the first table show the raw distributions of the comorbidities regarding the two groups. Furthermore, comorbidities were taken into account when the multiple regression analysis was executed, where all comorbidities were among the explanatory variables. Self-reported co-morbidities were the following in the analysis: Cardiovascular or cerebrovascular disease(s), Musculoskeletal disorder(s) and Gastrointestinal disease(s). Descriptive statistics regarding comorbidities can be read in lines 211-219.

Minor comments

Lines 38-39 is to replace after line 41.

The Introduction section has been enriched with several new references. The new structure is in line with the Reviewer’s comments.

Line 47 explain why the effectiveness can be lower than the expected.

The sentence was modified and restructured.

A reference for the sentence “The expected price is for an out-of-pocket influenza vaccine is 51 between $20 and $70” is to added.

We modified the sentence with new data to make this clearer and new references were also added.

After reference 10, line 56, you have to begin a new line.

We have made the requested change mentioned above which is now in line with the Reviewer’s suggestion.

The sentence 59-61 is to rewrite as it is not very clear.

The sentence was restructured to enhance readability and comprehensibility.

From line 63 the authors should add proper references.

Done, the sentence was enriched with more references.

The sentence at line 701 could be deleted.

We are unsure as to the meaning of this comment and ask the reviewer to kindly elaborate since the uploaded document contained 405 rows.

The study aim is not clear, explain the rationale to compare influenza vaccination coverage between self reported diabetic and non diabetic when non diabetic could suffer other co-morbidities.

The main focus of the manuscript is the influenza vaccination coverage of patients with self-reported diabetes. This has very important public health implications, which are described in detail in the introduction section.

The presence of different comorbidities can indeed have a significant confounding effect when comparing groups, however, the executed multiple logistic regression was adjusted for comorbidities and other potential confounders. Because of that method it is more clear which factors might be associated with vaccine uptake.

We believe that the Introduction section do describe the importance of vaccination among people with diabetes, which rationales the study.

The authors can expand the study limits:

It was noted in the Strengths and limitations section that due to the questionnaire; our results may under-represent reality. However, we modified the text to make it clearer. Now it can be read:

“However, using self-report-based questionnaires (e.g., the registered co-morbidities, such as diabetes) may have resulted in under-representation in our results as well as the misclassification as non-diabetic or diabetic respondents unaware of their condition, so it is important to consider these factors.”

Undiagnosed diabetes among individuals who respond not to be diabetics. It is well known that a considerable percentage of population does not know to be diabetic.

We would like to express our thanks for this comment. We modified the sentence to make this clearer, kindly see our response to the previous comment.

Round 2

Reviewer 1 Report (New Reviewer)

Authors have incorporated the suggestive changes which has improved the manuscript a big deal.

Reviewer 3 Report (New Reviewer)

The revised manuscript can be accept for publication.

This manuscript is a resubmission of an earlier submission. The following is a list of the peer review reports and author responses from that submission.

Round 1

Reviewer 1 Report

Overall, the manuscript offers a clear reporting of datasets. However, do consider the suggested comments below:

More description is needed for the European Health Interview Survey (EHIS). Why it is being performed, what would the survey wishes to achieve/expect? Either include this in the introduction or briefly expand it from the conclusion.

Lane 122- indicate the selection/factors that influenced the "data cleaning".

Lanes 143 and 145- define 'good' or 'bad' self-reported status.

Table 1- the gap of age group was too big (18-64). Is it possible to separate them into young adult vs middle-aged vs old-age to provide a more robust perspective?

Please improve references, i.e. minimise citations from internet which may be changed/obsolete after a certain period of time. Journal article citations are highly recommended. 

Author Response

Reviewer 1

Overall, the manuscript offers a clear reporting of datasets. However, do consider the suggested comments below:

More description is needed for the European Health Interview Survey (EHIS). Why it is being performed, what would the survey wishes to achieve/expect? Either include this in the introduction or briefly expand it from the conclusion.

Dear Reviewer, we are grateful for the comments and would like to thank you for giving us the opportunity to submit a revised draft of our manuscript.

We propose to address the first issue by rephrasing a sentence by adding extra information regarding EHIS:

“The data were obtained from the 2009, 2014, and 2019 Hungarian implementations of the cross-sectional EHIS, which was conducted in order to establish reliable health indicators in the EU Member States. With the survey it is possible to estimate the population's health status, lifestyle characteristics, self-care limitations, physical activity, nutrition, health risk behaviours and health care utilisation and satisfaction.”

Lane 122- indicate the selection/factors that influenced the "data cleaning".

We rephrased the sentence with more information regarding data cleaning. Now it can be read:

“The initial sample size was 16 480. After data cleaning, which involved the exclusion of those respondents who did not have data regarding diabetes status and were below 18 years, the total sample size was reduced to 15 874, with 4 899 respondents from the 2009 dataset, 5 620 respondents from the 2014 dataset, and 5 355 participants from the 2019 dataset.”

Lanes 143 and 145- define 'good' or 'bad' self-reported status.

We rephrased the affected sentences for better consistency:

The independent variables were sex (male/female), age group (18–64 years/65–X) and education level (primary/secondary/tertiary). Answers related to self-reported health status were the following: very bad, bad, and fair, which were pooled as ‘bad’; and good and very good, which were pooled as ‘good’ self-reported health status.

Table 1- the gap of age group was too big (18-64). Is it possible to separate them into young adult vs middle-aged vs old-age to provide a more robust perspective?

The different EHIS databases contained data regarding age-groups in different terms, so after merging the three datasets the following age groups were ready for pooling and analysing together: 15-17; 18-34; 35-64; 65-X. The focus of the study was the adult diabetic population of Hungary; therefore, all records of persons aged 15-17 years were deleted in the data cleaning phase. As the influenza vaccine is available free of charge for the elderly population in Hungary and in most European Union Member States, it was considered best to contrast the age groups 18-64 years versus 65 years and over. In addition, we performed the analysis with detailed information regarding the age-groups, but the age-groups of 18-34 and 35-64 were considered as homogeneous during the analysis, that is why we pooled them together in the final analysis. We believe that this comparison describes the Hungarian situation best.

Please improve references, i.e. minimise citations from internet which may be changed/obsolete after a certain period of time. Journal article citations are highly recommended.

In the manuscript, ten citations were cited from the internet. The text is now enriched with several references in order to provide additional support and information next to online citations. The online references which are available on the internet were not deleted, as in many cases they contained reliable data that best describes the current situation or had not been published elsewhere in an appropriate format; however, published references from different journals were added next to most online citations.

Reviewer 2 Report

In this paper, the authors present an analysis of the factors associated with influenza vaccine uptake, with a special focus on diabetes, in three iterations of the Hungarian implementation of the European Health Interview Survey (2009, 2014, 2019). Overall, they report that influenza vaccination rates are well below the European Union target of 75% of the general population, although the rates are slightly higher among diabetic vs non-diabetic individuals. Age, education and sex were associated with vaccine uptake. The authors conclude that a concerted effort is needed to promote influenza vaccine uptake, especially among diabetic patients, who are at increased risk of severe flu disease and complications, and who would get the vaccine free of charge in the Hungarian system.

Overall, the results are interesting, if not particularly novel. I think the presentation is somewhat complicated when presented by strata, and it makes it hard to pick out a single main message.

I think that a propensity score matched analysis would be more appropriate (propensity to get vaccinated; matching diabetics with non-diabetics). This would make it possible to match for the confounders in one single analysis. The authors should try this approach, to see whether a more uniform message would emerge.

Author Response

Dear Reviewer, we are grateful for giving us the opportunity to submit a revised draft of our manuscript.

We agree with the reviewer that a study with the objective of assessing whether diabetics are more likely to be vaccinated than non-diabetics could benefit from an analytic approach based on propensity score matching. However, this was not the objective of the study – we looked at vaccination coverage and its influencing factors among self-reported diabetic and non-diabetic respondents, that is why a multiple logistic regression model with interactions was performed.

Reviewer 3 Report

In this report, the authors leveraged European Health Interview Surveys to identify predictors of influenza vaccination status among Hungarian diabetic patients.

1.      One factor the authors investigated was the frequency of doctor visits, with a focus on visits within the prior year. They suggest that patients who visit their doctor on a yearly basis are more likely to be vaccinated. This would seem somewhat intuitive as I would think that doctors are inclined to suggest influenza vaccination, particularly to diabetic patients with co-morbidities. Were there any survey questions about healthcare workers readily suggesting vaccination? If not, do you have any insight into the prevalence of doctors promoting vaccination during visits?

2.      Related to point 1, I would also expect that patients who visit their doctor on a frequent basis may be more concerned with their health and/or have higher rates of co-morbidities. Would this factor into patients’ willingness to be vaccinated?

3.      The data suggests that the elderly are vaccinated more frequently. Is this an effect of the elderly being more susceptible to severe influenza disease? Would this correspond doctor recommendation of vaccination raised in point 1?

4.      In line 267, the authors state that sex differences may contribute to vaccine acceptance. Please elaborate on this point with specific examples.

5.      Please expand on how you intend to use the generated data to improve vaccination rate. I feel this should be a key aspect of the discussion as it is related to the primary research objective.

6.      Overall, the data presented appears to be more correlative than explanatory. However, the abstract states that the primary objective of this study was to identify characteristics that influence vaccination status. Perhaps a bit more explanation on how frequent doctor visits and education impact vaccination status would be helpful.

Author Response

Reviewer 3

In this report, the authors leveraged European Health Interview Surveys to identify predictors of influenza vaccination status among Hungarian diabetic patients.

Dear Reviewer, we are grateful for the comments and would like to thank you for giving us the opportunity to submit a revised draft of our manuscript.

  1. One factor the authors investigated was the frequency of doctor visits, with a focus on visits within the prior year. They suggest that patients who visit their doctor on a yearly basis are more likely to be vaccinated. This would seem somewhat intuitive as I would think that doctors are inclined to suggest influenza vaccination, particularly to diabetic patients with co-morbidities. Were there any survey questions about healthcare workers readily suggesting vaccination? If not, do you have any insight into the prevalence of doctors promoting vaccination during visits?

The primary objectives of the EHIS were not vaccination and factors influencing vaccination, but to establish reliable estimations about health status indicators of the adult population from private households; therefore, there were no extra questions or data towards vaccination or vaccination support.

According to the literature, it seems that recommendations done by the care staff could help to increase vaccine acceptance, and with it, vaccination prevalence. Indeed, vaccination is required annually, but vaccine hesitancy is still an ongoing issue not just in the general population but among those working in the healthcare sector.

Prematunge C, Corace K, McCarthy A, Nair RC, Pugsley R, Garber G. Factors influencing pandemic influenza vaccination of healthcare workers--a systematic review. Vaccine. 2012 Jul 6;30(32):4733-43. doi: 10.1016/j.vaccine.2012.05.018. Epub 2012 May 27. PMID: 22643216.

Hollmeyer HG, Hayden F, Poland G, Buchholz U. Influenza vaccination of health care workers in hospitals--a review of studies on attitudes and predictors. Vaccine. 2009 Jun 19;27(30):3935-44. doi: 10.1016/j.vaccine.2009.03.056. Epub 2009 Apr 8. PMID: 19467744.

Murray E, Bieniek K, Del Aguila M, Egodage S, Litzinger S, Mazouz A, Mills H, Liska J. Impact of pharmacy intervention on influenza vaccination acceptance: a systematic literature review and meta-analysis. Int J Clin Pharm. 2021 Oct;43(5):1163-1172. doi: 10.1007/s11096-021-01250-1. Epub 2021 May 28. PMID: 34047881; PMCID: PMC8161720.

However, we do believe that most doctors do recommend vaccination for their patients, but unfortunately there are no exact Hungarian data regarding the prevalence of doctors promoting vaccination during visits, only about knowledge, motivation, and attitudes of Hungarian doctors towards vaccination or the willingness to be vaccinated by pharmacists.

Rurik I, Langmár Z, Márton H, Kovács E, Szigethy E, Ilyés I. Knowledge, motivation, and attitudes of Hungarian family physicians toward pandemic influenza vaccination in the 2009/10 influenza season: questionnaire study. Croat Med J. 2011 Apr 15;52(2):134-40. doi: 10.3325/cmj.2011.52.134. PMID: 21495195; PMCID: PMC3081211.

Galistiani GF, Matuz M, Matuszka N, Doró P, Schváb K, Engi Z, BenkÅ‘ R. Determinants of influenza vaccine uptake and willingness to be vaccinated by pharmacists among the active adult population in Hungary: a cross-sectional exploratory study. BMC Public Health. 2021 Mar 17;21(1):521. doi: 10.1186/s12889-021-10572-8. PMID: 33731073; PMCID: PMC7967972.

  1. Related to point 1, I would also expect that patients who visit their doctor on a frequent basis may be more concerned with their health and/or have higher rates of co-morbidities. Would this factor into patients’ willingness to be vaccinated?

We adjusted the analysis for self-perceived health status and co-morbidities as well. The effects of the doctor visit frequency variable are thus unconfounded by these variables, the effects of which are consistent with the notion that healthier people tend to somewhat downplay the importance of vaccination.

  1. The data suggests that the elderly are vaccinated more frequently. Is this an effect of the elderly being more susceptible to severe influenza disease? Would this correspond doctor recommendation of vaccination raised in point 1?

Many factors could contribute to the vaccination of the elderly. On the basis of the present analysis and in the absence of more relevant data, it is not possible to give a clear and quantifiable answer for this question. Indeed, the elderly population is at increased risk regarding influenza, as they have an increased risk of morbidity and mortality during influenza epidemics. The presence of co-morbidities contributes to this association, making them a prime target group for vaccination. Therefore, elderly people with more frequent doctor visits might receive more frequent vaccine recommendations. We also concluded this in the Discussion, whereas thy can be read:

“The elderly population had higher a proportion of vaccinated members, and thus more willingness for vaccine uptake; moreover, these individuals are in the care of healthcare providers. (23-25) This does not mean that the older age increases vaccination coverage, but it does highlight the fact that it would be useful to identify the factors that contribute significantly to vaccination uptake among elderly people. Therefore, it would be important to examine whether public health interventions could be taken to further increase vaccination coverage among the elderly and to raise vaccination coverage in younger age groups to a level similar to that of the elderly. “

However, assessing the effect of older age on vaccination was not our main intention, therefore it was not within the scope of the present study.

  1. In line 267, the authors state that sex differences may contribute to vaccine acceptance. Please elaborate on this point with specific examples.

It is difficult to provide examples with which to go beyond a detailed description of between-sexes differences in vaccination across strata formed by diabetes and frequency of doctor visits, which we did provide in the manuscript. The EHIS response data simply do not carry the information necessary to support any such elaboration. However, detailed description across different strata can be seen in the manuscript:

“Male vs female differences were observed in the terms of doctor visits and vaccine uptake, indicating that diabetic men with more frequent doctor visits were more willing to be vaccinated, and diabetic men who did not visit their doctor regularly had lower odds of being vaccinated. In contrast, non-diabetic women who did not visit their doctor within one year had in-creased odds of being vaccinated. This could indicate that differences between males and females may contribute to vaccine acceptance.”

  1. Please expand on how you intend to use the generated data to improve vaccination rate. I feel this should be a key aspect of the discussion as it is related to the primary research objective.

The main aim of this study was to assess vaccination coverage and its influencing factors among self-reported diabetic and non-diabetic respondents. Therefore, the authors primary intention was to explore those factors which could contribute to vaccination. We believe that this study can contribute to a better understanding of the factors behind low vaccination coverage observed among self-reported diabetic people in Hungary. It is important to emphasise that not only the elderly and elderly diabetics should be in the focus of care, but it is important to raise the vaccination coverage of other target groups at least to the level of the elderly, as it can be seen in the manuscript.

In addition, the present study provides insights into which target groups of diabetics have lower vaccination coverage and which target groups are the least likely to be vaccinated, thus making themselves at increased risk of influenza. Furthermore, it is highlighted in the manuscript that public health authorities and healthcare practitioners should be aware of the potential effects of influenza, further aggravating the fact, that there are no well-established monitoring indicators regarding influenza vaccination among diabetic people, so the net health gain due to vaccination cannot be measured directly.

  1. Overall, the data presented appears to be more correlative than explanatory. However, the abstract states that the primary objective of this study was to identify characteristics that influence vaccination status. Perhaps a bit more explanation on how frequent doctor visits and education impact vaccination status would be helpful.

We agree with the Reviewer, that is why we added an extra sentence to the Discussion. Thy can be read:

“Furthermore, frequent doctor visits might be associated with greater exposure to vaccination advice; higher education might be associated with higher levels of health literacy, which in turn might be associated with better vaccine acceptance.”

Round 2

Reviewer 2 Report

Thank you for your response, although I note that my suggestion was not taken on board. 

Reviewer 3 Report

I thank the authors for providing responses to my points. It seems to me that the survey used in this study provides topline responses, but may not allow for a more in-depth analysis. In my opinion, this remains a knowledge gap, however, I am unsure of the approach for addressing this.